# Crisis leadership and strategic decisions in Swedish maternity care during the COVID-19 pandemic: A deductive analysis from the COPE staff project

Emelie Stotzer[1,2]*, Sofie Graner[3,4], Verena Sengpiel[5,6], Anna Wessberg[5,7], Magnus Akerstrom[1,8], Karolina Linden[7]

1 School of Public Health and Community Medicine, Institute of Medicine, Sahlgrenska Academy, University of Gothenburg, Gothenburg, Sweden, 2 Department of Gynecology, Angered Hospital, SV Hospital Group", Gothenburg, Region Västra Götaland, Sweden, 3 BB Stockholm, Danderyd University Hospital, Stockholm, Sweden, 4 Department of Medicine, Centre for Pharmacoepidemiology, Karolinska Institute, Stockholm, Sweden, 5 Department of Obstetrics and Gynecology, Sahlgrenska University Hospital, Gothenburg, Region Västra Götaland, Sweden, 6 Department of Obstetrics and Gynecology, Institute for Clinical Sciences, Sahlgrenska Academy, University of Gothenburg, Gothenburg, Sweden, 7 Institute of Health and Care Sciences, Sahlgrenska Academy, University of Gothenburg, Gothenburg, Sweden, 8 Region Västra Götaland, Institute of Stress Medicine, Gothenburg, Sweden

ཐ These authors contributed equally to this work.
* emelie.stotzer@gu.se

## Abstract

### Background

Healthcare managers played a crucial role during the COVID-19 pandemic, tasked with organizing care for a new medical condition, implementing restrictions to reduce infection spread, and handling unprecedented staff shortage while maintaining operability and ensuring a sustainable working environment for employees. The aim of this study was to generate knowledge about Swedish maternity care managers' decision-making, by exploring their crisis management to cope and to mitigate the pandemic's effects in working units.

### Methods

Semi-structured interviews were conducted with 18 managers from various organizational levels at different Swedish maternity care units during the third wave of the pandemic (March – June 2021). A deductive qualitative content analysis informed by the Consolidated Framework for Implementation Research (CFIR), was performed.

### Results

The pandemic compelled managers to employ innovation, reorganization and altered working routines to address external and internal demands. Many decisions were made rapidly, often with limited information and without sufficient time for thorough

**Data availability statement:** Relevant excerpts from the transcripts that support the findings of this study are available upon reasonable request, pending approval from the University of Gothenburg from Professor Karin Ahlberg (Head of Department), who serves as the institutional representative for data access matters, at karin.ahlberg@gu.se. Due to the highly limited recruitment pool and the sensitive nature of the interview material, the full dataset contains potentially identifiable information that cannot be fully anonymized without compromising participant confidentiality and cannot be shared openly in compliance with the ethical approval granted for this study.

**Funding:** This research was funded by the Swedish Research Council for Health, Working Life and Welfare FORTE (Karolina Linden, 2022–00187), the Swedish Research Council FORMAS (Karolina Linden, 2020–02767), the Swedish government and county councils, and the ALF agreement (Karolina Linden, ALFGBG-1006131; Verena Sengpiel, ALFGBG-970689). The funders had no role in study design, data collection and analysis, decision to publish, or preparation of the manuscript.

**Competing interests:** The authors declare that they have no competing interest.

consideration. Peer support and effective communication were identified as essential for coping with the situation. Managers worked long hours and expressed both challenges in their decision-making and pride in their ability to fulfill their roles. Transparency in the decision-making process, along with continuous reflection and evaluation, were viewed as successful crisis management tools for enhancing workplace sense-making and helping employees maintain motivation.

## Conclusions

Throughout the pandemic, managers had to develop their own methods for making and implementing decisions, aimed at ensuring patient and employee safety and well-being, often without organizational guidance on leading in crisis. It is essential to share knowledge about effective regulatory strategies to mitigate crisis impacts and to incorporate these strategies into crisis management frameworks to strengthen preparedness for future emergencies.

## Introduction

Maintaining both the organisational functionality and employee well-being during a crisis requires effective crisis management [1]. Such management typically involves changes in decision-making, policies, and procedures within an organisation [2]. Managers play a crucial role in this process [3], particularly during crises like the COVID-19 pandemic, where they were compelled to make drastic and often difficult decisions with limited information [4,5].

Crisis management involves managers at various organisational levels. Top management must provide essential information about the crisis and specify necessary measures, while middle and front-line managers are responsible for implementing these measures and mediating sense-making between upper management and frontline employees [6]. A key aspect of effective crisis management is systematically learning from past crisis to transfer lessons to future incidents [1].

Although managers of maternity units deal with unplanned events on a regular basis, leading through persistent crises like the COVID-19 pandemic has not previously been explored, in this specific speciality/clinical discipline. And only few studies report experiences from health care managers in general, in the Swedish setting/ health care sector during the Covid-19 pandemic [4].

In maternity care, maintaining a functioning service during the pandemic was critical, as this care cannot be postponed and life-threatening emergencies can arise at any time [7–10]. Pregnant, both patients and coworkers, were identified early on as a risk group. Unlike many countries, Sweden did not implement an official lockdown during the pandemic; instead, it introduced recommendations for social distancing, refraining from nonessential travel, and working from home. This unique situation exposed Swedish healthcare managers to unprecedented challenges [11]. Increased workloads and negative health effects for maternity care workers has been reported [8,12–17]. However, there is limited information on the managers' situation and their

regulatory strategies, particularly how they managed decision-making while striving to maintain organisational functionality and employee well-being [4,17].

Multiple factors related to organisational context and the implementation of measures can significantly influence the outcomes of workplace interventions, including crisis management. This understanding has led to the development of several frameworks that identify key factors at various levels promoting or hindering effective implementation. One widely used framework is the Consolidated Framework for Implementation Research, CFIR [18], which facilitates evaluation and learning processes within the workplace.

Thus, the aim of the study was to generate knowledge on how Swedish maternity care managers navigated decision-making under pandemic uncertainty by evaluating the implementation of crisis management.

The study entailed interviewing managers at different organisational levels, exploring how managerial decisions were made and implemented.

## Methods

The number of births in Sweden is approximately 115 000 each year, occurring in about 45 publicly financed maternity-care units, located in 21 healthcare regions. Data was collected by individual digital interviews during the third wave of the pandemic, March – June 2021 [19]. This work is part of a Swedish national multi-centre study within the Swedish maternal health care, the COPE Staff study, previously described in detail [13].

This study was approved by the Swedish Ethical Review Authority on August 27, 2020 (approval number 2020−03446), with amendments on November 12, 2020 (2020−05747) and March 1, 2021 (2021−00854). The participants agreed to participate after giving their informed consent.

The research team has broad experience of clinical work within maternity care and during the pandemic, which had to be considered during analysis. ES, SG and VS are obstetricians and AW and KL are midwives. Further, SG is the head of operations at a birth unit, AW has worked as a unit manager, ES has experience of union work and MA has a background in occupational medicine, VS was medically responsible for a special antenatal care unit during the pandemic and part of the medical reference group at the maternity unit. The interviews were performed by a research assistant with no previous experience of work in health care and no relation to the informants.

Eighteen managers at different levels within the organisation, from ten hospitals with a birth unit in eight regions, (S1 Table). We applied a purposeful sampling strategy with the aim of achieving maximum variation. Managers were strategically selected to represent different organisational levels (first-line, middle, and senior management), both university and non-university hospitals, and geographical spread across the country. We also sought to include clinics that had made different operational choices during the pandemic (e.g., whether to allow partners on postnatal wards or not) and managers with varying lengths of leadership experience. In some cases, multiple managers from the same organisation were included in order to capture variation within local decision-making structures. The sampling strategy was informed by the concept of "information power" [20,21] emphasizing that the richness and relevance of the data, rather than sample size alone, determine the adequacy of the material for the study aim.

The trained research assistant, conducted the interviews, based on a modified version of the CFIR interview guide [22] (S1 File – Interview Guide). A test interview was conducted to evaluate and perfect the adaptation of the questions. The opening question was: *Can you tell me about an operational decision you made during the pandemic which was crucial for the clinic or demanding for you as a person?* All interviews were audiotaped and transcribed verbatim. Interviews lasted between 34 and 70 minutes (mean 51 minutes). The test interview was not included in the analysis.

The CFIR 2.0 is a determinant framework that combines concepts drawn from multiple implementation frameworks, models, and theories. It organizes 48 constructs across five domains: 1) Innovation domain (what is implemented); 2) Outer setting domain (i.e., governmental or regulatory decisions, hospital systems etc.); 3) Inner setting domain (i.e., the internal organisation); 4) Individuals domain (i.e., roles and characteristics of individuals); and 5) Implementation process

domain (i.e., activities, strategies and documentation of implemented decisions). The framework can be applied in a flexible way, and all constructs will not necessarily be present in the data [18] (S2 Table).

In our analysis we have defined the crisis management amongst the managers as the innovation, consisting of various measures such as decision-making, implementation of changes, and ways of managing one's own well-being. Consistent with CFIR 2.0, the framework was used as an analytic structure to organize and interpret managers' accounts of crisis management practices, rather than to evaluate the effectiveness or evidence base of a predefined intervention.

The choice of CFIR was based on its proven flexibility across study design, data collection, and analysis, as highlighted in a systematic review by Kirk et al. [23]. During the pandemic, crisis management within maternity care could not be planned in a structured way. Instead, the managers were instructed to perform a context-adapted crisis management process, including designing and implementing context-specific preventive measures to maintain operability and ensure a sustainable working environment for employees, in accordance with recommendations for organizational-level workplace interventions [24]. Using a theoretically grounded framework made it possible to capture important dimensions of the implementation process that might not otherwise have been articulated, thereby adding depth and nuance that a purely inductive analysis likely would not have achieved.

A deductive content analysis, inspired by Elo & Kyngäs [25], was performed informed by the CFIR [18]. The analysis followed the recommended phases of preparation, organisation, and reporting. All transcripts were read multiple times by the research team to achieve immersion in the data. A coding framework based on CFIR domains and constructs was developed and refined iteratively. Each interview was coded manually by at least two researchers (ES, SG, AW, KL). A subset of interviews was first coded jointly to establish a common understanding of the framework and to pilot the coding procedure. Discrepancies in coding were documented and resolved through discussion in iterative team meetings until agreement was reached. When consensus could not be reached immediately, transcripts and codes were revisited, and the coding framework was adjusted accordingly. This process ensured that interpretations were not determined by a single researcher's perspective. Rigour was enhanced through systematic consensus-building and continuous reflexive discussions within the multidisciplinary team. This iterative and collaborative approach strengthened the credibility and dependability of the findings, in line with recommendations for ensuring trustworthiness in deductive content analysis [26].

Meaning units were coded into the most relevant CFIR domain and then grouped under constructs. Codes were further consolidated into higher-order categories to capture patterns across managers' accounts. (S3 Table)) illustrates an example of the coding process, from meaning unit to code, construct, and domain, and reflects the systematic steps undertaken to ensure transparency. Rather than emphasising frequency counts, which are not central to the study aim, the analysis focused on the richness and variation of the data, in line with qualitative research principles

## Results

The analysis identified 33 (out of 48) constructs within the five domains of the CFIR framework.

### Innovation domain

The pandemic demanded quick innovations, reorganisations and changed work practices. These occurred both in direct patient care, as well as in healthcare governance and decision pathways, since new risks and care needs were introduced frequently. Numerous decisions of short duration were taken. A way of managing this was to sketch different scenarios with various levels of infection spread and preventive measures. The managers were used to changing routines and practice but normally had some kind of medical evidence to support their decisions. Fear of spreading the infection led to separating infected and non-infected patients, prohibiting relatives from visiting, cancelling physical meetings, and catalysed innovations such as new models of communication, foremost of which was in digitalisation.

*"We didn't have a long and proven experience to lean on. It was really 'how do we think about this? How do we protect...? What should we do? How do we implement...?' Otherwise, you always have guidelines, instructions, and protocols to lean on, but they didn't exist in this situation."*

Some clinics made a new capacity unit to facilitate this work, a task force made up of a small group of trusted colleagues with specific skills, knowledge, or networks. This decision was justified by the need for additional expertise within the management team and the need for rapid decisions.

*"Actually, no other knowledge base than what concerned management and control in an emergency situation, where we assessed that it was not effective to have a large group of people, but we needed to have a small decision-making group that we could mobilize."*

Other clinics decided to utilize their existing management team and keep working according to their established routines. Regardless, the need for competent and dependable colleagues was empathised, peer support was necessary for coping

*"We had close contact with the union. In fact, we had cooperation meetings every week because... We realized that if we involve the union […] and give them some advance information, we can also test their thoughts and understand what their members think and feel."*

**Outer setting domain**

The managers experienced maternity care being exposed to tangible external pressure coming from health authorities, politicians, patients, and their families. For instance, media revealed different routines regarding accompanying partners. In addition, strong political pressure to remove the prohibition of companions in postnatal care was put on the managers within some regions, as the public opinion of dissatisfied patients became more commonly known.

*"...my colleagues and I, fellow heads of department, felt a lot of pressure from the politicians to loosen these visitation restrictions because they, in turn, surely faced a lot of pressure from the public... Even though they said they wouldn't interfere with the medical aspect, they did"*

The managers used partnerships and connections to gain new knowledge about how to handle the risks of the virus. It was hard for them to keep track of new knowledge. To keep informed they connected with peers through social media and digital meetings, received information from their professional associations, and tried to keep up with recommendations from the Public Health Agency.

*"Swedish Society of Obstetrics and Gynaecology arranged webinars once a week where the current state of knowledge was reviewed, and it was largely about pregnancy. It also covered general considerations before surgery. "*

In contrast to ordinary operations, economic consequences following decisions taken posed very few constraints. This relief eased decision making since finances did not need to be prioritised as usual.

*"We have been given permission […] to book our COVID-related expenses to a specific account […] And it includes both personnel costs and materials, as well as medications that we use increasingly due to COVID. […] It's a lot of money."*

## Inner setting domain

Pre-pandemic capability and capacity differed between the participating clinics. Development of digital solutions, employee's access to their own computers, information flow, decision culture, care ideology, and support to and experience of leadership among the informants affected both the managers' decision making and the ability to manage their operations. Other examples are single or shared rooms in the postnatal ward and established home visits as part of postpartum care.

It was common practice at most hospitals to re-assign staff from all specialities to help with COVID-19 infected patients. The managers expressed gratitude that maternity care was spared since operations could not function without specialised staff.

*" No one wants an orthopaedic surgeon to deliver a baby, for example, nor can a midwife be replaced by a nurse from another department."*

Managers faced staff shortages daily both due to staff with diagnosed COVID-19 and suspect symptoms, and staff belonging to a high-risk group for severe COVID-19, who had to be reassigned to different tasks. Managers adopted different approaches when it came to sparing staff from working directly with infected patients. Some strictly adhered to guidelines set by the authorities.

*" We have […] requested a medical certificate from their doctor, stating that they belong to a high-risk group and should not work with COVID-positive or suspected COVID-positive patients. "*

Others decided to go further to protect their pregnant staff. *" It was a decision I made against the recommendation, that we have tried to keep our pregnant staff away from COVID care from the beginning. […] Our reasoning was that since we don't know, it's better to be safe than to expose them to any risk. "*

Some managers expressed frustration over a perceived inflexible or rigid organisation in terms of moving staff internally. A common example was re-assignment of pregnant staff after the authorities declared them to belong to a risk group there was no easy way to facilitate a transfer to other parts of the health-care organisation.

A part of the organisational culture that was immensely helpful for managers was a collaborative spirit and improved collaboration with other units, decreasing the sense of rivalry between departments focusing on all patients' needs.

*"Collaboration. Everyone has helped each other. It's somehow an external enemy that just has to be fought against. There has been a sense of togetherness that, while not entirely new to us, has become stronger. However, it is clear that the long duration of this situation is taking its toll. So, the current challenge is finding that endurance energy. "*

## Individual's domain

The capability of the interviewed managers to handle the situation related to experience of the role, time in the position, and experience within the discipline and research. *"This specialty is one of the most acute specialties out there. We're used to detecting danger and acting quickly; that's what we do at night. And we like that, we thrive on it, that's why we work [laughter] in this field, so to speak. So our personality is suited for [laughter] crises. But short crises, long crises, none of them are fun"*

Also, the understanding of what is a good leader is revealed in the managers' evaluations. *"So, generally, it's the importance of daring to make decisions. You can make the wrong decisions, but it's worse not to make any decisions at all"*

Informants expressed feelings of being proud, being needed and making a difference in a complex and demanding situation motivating them to continue despite the extreme workload. *"Sometimes I actually think, "Imagine that I got to be a part of this." A year ago, I never thought I would have to be involved in managing obstetrics for women during a pandemic.*

*And we have done it well. "* Managers reflected on their own moral compass in difficult decisions. *"The absolutely biggest effort required when it comes to personnel, it's those assigned to epidemic duties, those who had to assist in various ways with COVID patients... the most demanding job, in my opinion, as a manager... it's like sending out troops to war as a general. You try to convince them that this is very important... and you don't know if everyone will come back... We didn't know how dangerous this virus was. "*

### Implementation process domain

The pandemic created a sense of urgency and decisions needed to be made ad hoc and hence tailored and reevaluated frequently. One of the greatest challenges was to keep all staff informed about the latest updates and to keep themselves informed about what was happening on the floor. Various information strategies were launched, from daily e-mail updates to digital meetings and weekly newsletters. Some managers adapted new strategies, such as using appointed colleagues to evaluate and assist information flow to and from the managers.

*"We appointed two representatives within the staff group to act as COVID ambassadors. They were out among the staff to ask questions and serve as the link between us."*

Another challenge was how rapidly decisions had to be updated and the fact that they might communicate something that was outdated in just a few days' time.

*"This decision is made based on what we know now and what we can deliver at this moment. We decide on a concept and communicate it. That's how it has to work for now. But in two weeks, or a week, or even three days, we may know more. The Public Health Agency may come up with something new, and then we'll have to say, "Now we're doing it this way instead."*

A challenge was to maintain engagement throughout the whole pandemic. A psychological blunting was reported. *"One is tired of the fast flow of information, tired of the rapid changes. They're a bit tired... [In our clinic] people are also tired that... now patients are starting to lose... they don't realize it's a pandemic, while we're still living in it. "*
Many of the changed routines and necessary priorities created a negative atmosphere among the employees. The standard of care was perceived as lower than that desired by the employees, creating ethical stress. *"Some have voted with their feet and quit. That's actually the case. Even the former head of the maternity ward got tired and left. We implemented an entirely new management team. They have worked wonders, putting in tremendous effort to motivate the staff and also gather ideas from them."*
The managers constantly needed to reflect on and evaluate their decisions. Peer support was perceived as a valuable tool. This occurred through both formal and informal management groups, as well as through digital contact with various authorities, specialist associations, and colleagues at departments of obstetrics and gynaecology in different parts of the country.

*"We have summarized the pandemic a few times at the clinic, noting both strengths and concerns. One of the strengths is that there is even less silo thinking now. It's very clear that there is more collaboration. Additionally, transparency has been necessary to prioritize health care effectively"*

### Discussion

The COVID-19 pandemic posed significant challenges for both operations and employees within maternity care units in Sweden [9,12]. However, little attention has been given to the working conditions and practical strategies employed by managers during a crisis [5]. This study investigates the decisions-making process of managers during the COVID-19

pandemic, with the aim of generating knowledge about their crisis management strategies for mitigating the pandemic's impact on operations and employee well-being.

Our findings indicate that managers employed innovative approaches, reorganizations, and adjusted work routines to address both external and internal demands during the pandemic. These circumstances often required swift decisions to be made with limited information, resulting in short-lived solutions. Transparency in decision-making and peer-support, particularly through ongoing reflection and evaluation, emerged as key strategies for enhancing sensemaking in the workplace, which in turn helped maintain employee motivation. While effective communication was recognized as a critical regulatory strategy, it proved to be challenging in practice. Managers expressed both the difficulties they faced in decision-making and a sense of pride in their ability to fulfil their leadership roles. Transferring this generated knowledge may enhance organizations' capacity to navigate future crises successfully.

Ensuring sustainable working conditions for managers during a crisis is a crucial component of crises management, as leadership is often held accountable for maintaining operations and securing a healthy work environment for employees, especially when healthcare systems are under strain [4]. Our results suggest that managers faced increased demands during the COVID-19 pandemic, due to external pressures to reduce the spread of the virus, accompanied with national and local restrictions. They also had to address dissatisfaction with these measures among predominantly external actors such as partners, media and the general population. Communicating changes to work routines, ensuring adequate staffing, and managing employee worry further heightened these demands. However, increased demands do not necessarily result in adverse working conditions for managers, as access to adequate resources can mitigate the impact of heightened demands [25]. This was also seen in our material, i.e., expressed by pride in their role.

One valuable resource identified in this study was organizational clarity regarding the de-prioritization of financial considerations, which allowed managers to focus on mitigating actions without budgetary constraints. Additionally, stronger team cohesion, a sense of pride, and the feeling of being needed and performing meaningful work were recognized as important resources. Organizational capacity in digitalization was also identified as a resource when present, and a limitation when absent. Similar shifts in job demands and resources have also been observed among employees in maternity care [9,13–18,27] and among healthcare managers in other sectors [5].

Examining the balance between the demands placed on managers and the resources available to them during crisis leadership revealed a potential lack of organizational support. Managers were required to invest considerable time being physically present on site, both during regular working hours and outside office hours, to mitigate the negative effects on operations and employee well-being during the pandemic. Therefore, leveraging the lessons learned from the pandemic to identify organizational-level interventions that can be integrated into future crises management plans is crucial (see below). Doing so will not only enhance the organization's resilience during periods of strain (2) but also improve working conditions for managers [28].

This study further illuminates the specific tasks undertaken by managers during a crisis and the regulatory strategies they employed to mitigate the impact of crises on operations and employee well-being. Our findings highlight the necessity for rapid decision-making and the development of individualized coping strategies by managers, given the high levels of uncertainty. Additionally, managers faced the pedagogical challenge of continuously communicating shifting routines to their teams. These findings align with previous research on leadership during the COVID- 19-pandemic, which emphasized the need for leaders to act swiftly, filter and interpret complex information, and adapt services accordingly [8,29].

Managers were also tasked with addressing employees' varied expressions of worry and anxiety. Approaches to supporting employees differed among managers, with few reporting the availability of organizational support. Consequently, many managers had to independently devise strategies for offering employee support. Some adopted uniform directives for all employees, while others implemented a more individualized approach. Effective leadership during the COVID-19 -pandemic, as noted in earlier studies [8,29], involves not only providing emotional and social support but also ensuring the availability of psychosocial resources, such as, e.g., adequate information and other context-appropriate measures. Access to organizational support can alleviate some of the pressures faced by managers in such crises.

Regarding the regulatory strategies employed by managers, a key approach repeatedly mentioned was the ability to create their own support networks by informally or formally connecting with valuable colleagues. Why some clinics kept existing formal groups and others not, cannot be explained by our material. This collaboration, coupled with shared consensus and the best available experience, provided managers with a sense of security in frequently making and revising decisions as new information and experiences arose. Another regulatory strategy involved maintaining transparency in the decision-making process and continuously communicating updates on changed working routines, decisions, and COVID-19-specific knowledge. The practical execution of communication varied across organizations, influenced by their culture, practices, and access to communication channels, with most managers finding this task particularly demanding. Effective communication is critical during a crisis and first- and middle line managers play a crucial role in facilitating sense-making between top management and frontline employees [7].

Spending significant time physically present on-site both during and outside regular working hours, was deemed necessary. This strategy, alongside social support from coworkers, has been previously identified as beneficial [5]. However, reliance on individuals' personal time as a resource creates vulnerability within the organization. This approach highlights the absence of more systematic strategies to provide adequate resources, thereby balancing the demands placed on managers.

## Strengths and limitations

Our observation among managers is equivalent insignificant of experience and level in the organization. By interviewing managers, we gained insights into their perceptions of their leadership abilities and actions during the crises. The interviewed group included managers from various organizational levels, with responsibilities across different professions and varying levels of experience, offering a broad and diverse perspective. A key strength of this study is its contribution to the understanding of managers' demands and resources in their work environment, affecting their work-related health, which remains an area in need of further research. However, a limitation of the study is that it reflects managers' self-assessments of their performance, rather than measuring the impact of their leadership on the organization or their employees. This is an area that requires further investigation. The qualitative approach makes the generalization to other areas within the hospital difficult, however, findings may be transferred if contextual factors are carefully considered. It should also be noted that the CFIR framework has most commonly been used to guide planned and structured implementation of evidence-based methods, rather than context-adjusted organisational work-environment interventions. In this study, CFIR 2.0 was applied in a context-sensitive and partial way to structure managers' accounts of crisis management. The findings should therefore be interpreted with this specific application and context in mind.The research team's professional backgrounds inevitably shaped the research process. Several authors had extensive clinical and leadership experience within maternity care during the pandemic, which provided contextual sensitivity but also required reflexivity throughout the analysis. To strengthen trustworthiness, we applied strategies described by Elo et al. [26], focusing on credibility, dependability, confirmability, and transferability, adapted to a deductive content analysis approach. Credibility was enhanced by involving several researchers in the coding process, discussing and reaching consensus on how the CFIR framework was applied, and by illustrating the analytical process with examples. Dependability was supported by a transparent description of the data collection and analytic procedures, enabling other researchers to follow the logic of the study. Confirmability was strengthened by regular discussions within the research team, where different professional perspectives were used to challenge interpretations and ensure that findings were grounded in the data rather than in individual preconceptions. Transferability was addressed by providing a detailed description of the study context and sampling strategy and thereby allowing readers to judge the relevance for other settings. These strategies, in line with the SRQR standards [30], increase the trustworthiness of the findings and support the validity of the conclusions drawn.

A methodological contribution of this study is the application of CFIR in the context of crisis leadership [18,31]. While CFIR is commonly used to guide planned implementation processes, it has also been highlighted as valuable for structuring data collection and analysis in diverse study designs [23]. In our study, applying CFIR as a deductive framework

 

enabled a systematic exploration of factors influencing managerial decision-making at multiple levels of the healthcare system, spanning innovation, inner and outer setting, individuals, and implementation processes. This provided a structured lens that allowed us to identify not only evident but also less articulated barriers and facilitators that managers themselves may not have expressed without a guiding framework.

By anchoring the analysis in CFIR [18], we were able to move beyond a generic thematic description of managers' experiences and instead situate our findings in relation to well-defined constructs of implementation science. This methodological approach strengthens the transferability of our results, as it links context-specific observations of maternity care managers to a broader conceptual framework. In doing so, the study contributes to methodological development by demonstrating how CFIR can be applied in unplanned crisis situations, thereby offering insights into how implementation frameworks can advance understanding of leadership and decision-making under extraordinary conditions.

## Lessons for future crisis leadership recommendations

Leveraging lessons from the pandemic to identify organisational interventions is crucial for enhancing resilience during future crises and improving managers' working conditions [28,32]. Our findings suggest several actionable lessons for future preparedness.

*First*, leaders benefit from rapid yet revisable decision-making protocols that make explicit how decisions are updated as evidence shifts, alongside clear internal communication channels to cascade changes efficiently. This aligns with empirical work underscoring crisis communication competencies in health systems and the need for structured internal communication routines during COVID-19 [32–34]. Hospital leadership training should incorporate competencies in organizational restructuring to foster agility and responsiveness amid rapid change, using scenario-based modules informed by COVID-19 experiences. This includes revising decision-making structures, clarifying roles and communication channels, and identifying key individuals and competencies.

*Second*, establishing and maintaining peer-support infrastructures (both formal and informal) is critical. In our study, managers actively sought collegial consultation to stress-test decisions and share emerging practices. Evidence from reviews of support interventions for healthcare workers during pandemics indicates that structured emotional/peer support can mitigate distress and enhance perceived capacity to cope. Organizations should formalize peer-support networks to support crisis leadership. [35,36]

*Third*, develop clear internal communication channels accessible both from home and at work. Digital platforms (for coordination, information dissemination and remote meetings) should be embedded in routine operations to be readily scalable in crises. Reviews and scoping syntheses during and after the pandemic describe how information and communication technology, enabled solutions and telehealth supported remote coordination, monitoring and communication, capabilities that map directly onto the challenges described by managers in this study [37]. However, security issues need to be considered even during pandemics when quick solutions are needed [38].

*Fourth*, to address enduring post-pandemic pressures, notably burnout and staffing instability, organisations should pair rapid decision protocols with supports that protect leaders' well-being (e.g., access to peer networks, transparent governance, and realistic workloads). Some recent multi-year and regional analyses show burnout remains elevated relative to pre-pandemic baselines [39], and systematic reviews report high prevalence across settings [37], underscoring the need for sustained organisational measures rather than episodic crisis fixes.

*Finally*, resilience is strengthened when these practices are maintained in non-crisis periods. Our results highlight leaders' reliance on transparent communication, rapid feedback loops, and trusted colleagues; recent work on crisis leadership competencies similarly emphasises anticipatory planning, prioritisation, and swift decision capacity [31], competencies that can be cultivated before emergencies occur.

Together, these lessons translate our CFIR-informed analysis into practical recommendations: codify dynamic decision-updating and internal communication routines; institutionalise peer-support mechanisms; invest in interoperable

digital tools and training; and proactively resource leadership well-being to buffer ongoing workforce strain. These steps are actionable in everyday operations and scalable in future crises.

**Key recommendations**

I. **Formalise rapid and revisable decision-making protocols**, including clear communication routines and defined roles.

II. **Institutionalise peer-support systems** (both formal and informal) to strengthen crisis leadership capacity.

III. **Embed interoperable digital communication platforms** into routine operations, ensuring secure, accessible coordination in crises.

IV. **Safeguard managers' well-being** by combining crisis decision supports with sustained measures against burnout (e.g., realistic workloads, staffing stability, transparent governance).

## Conclusion

Throughout the pandemic, managers had to develop their own methods for making and implementing decisions, aimed at ensuring patient and employee safety and well-being, often without organizational guidance on leading in crisis. It is essential to share knowledge about effective regulatory strategies to mitigate crisis impacts and to incorporate these strategies into crisis management frameworks to strengthen preparedness for future emergencies.

## Supporting information

**S1 Table. Informants.**
(DOCX)

**S2 Table. Domains and constructs.**
(DOCX)

**S3 Table. Example of the Analytical Process.**
(DOCX)

**S1 File. Interview Guide.**
(DOCX)

## Acknowledgments

We want to thank the Swedish network for national clinical studies in Obstetrics and Gynaecology, SNAKS, (26) for supporting the COPE Staff study and to participating respondents.

## Author contributions

**Conceptualization:** Sofie Graner, Verena Sengpiel, Anna Wessberg, Magnus Akerstrom, Karolina Linden.

**Formal analysis:** Emelie Stotzer, Sofie Graner, Anna Wessberg, Karolina Linden.

**Funding acquisition:** Karolina Linden.

**Methodology:** Sofie Graner, Anna Wessberg, Karolina Linden.

**Project administration:** Magnus Akerstrom, Karolina Linden.

**Supervision:** Magnus Akerstrom, Karolina Linden.

**Validation:** Verena Sengpiel, Magnus Akerstrom.

**Writing – original draft:** Emelie Stotzer, Magnus Akerstrom, Karolina Linden.

**Writing – review & editing:** Emelie Stotzer, Sofie Graner, Verena Sengpiel, Anna Wessberg.

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
