## [Decision Letter · Decision Letter 0]

13 Jul 2025

PONE-D-25-04207Leading in crisis - Swedish maternity care managers’ experience of decision making during the COVID-19 pandemic: A deductive content analysis from the COPE Staff projectPLOS ONE

Dear Dr. Stotzer,

Thank you for submitting your manuscript to PLOS ONE. After careful consideration, we feel that it has merit but does not fully meet PLOS ONE’s publication criteria as it currently stands. Therefore, we invite you to submit a revised version of the manuscript that addresses the points raised during the review process.

We look forward to receiving your revised manuscript.

Kind regards,

Andreas Vilhelmsson, Ph.D

Academic Editor

PLOS ONE

Journal Requirements:

“FORMAS – the Swedish Research Council for Sustainability (2020-02767) – for funding the research”

4. In the online submission form, you indicated that [Data contains potentially identifying and sensitive information, and there are ethical restrictions to sharing the data file as it includes individual interviews with participants. For data request, please contact The Swedish Ethical Review Authority: registrator@etikprovning.se].

8. Please remove all personal information, ensure that the data shared are in accordance with participant consent, and re-upload a fully anonymized data set.

Reviewers' comments:

Reviewer's Responses to Questions

**Comments to the Author**

1. Is the manuscript technically sound, and do the data support the conclusions?

Reviewer #1: Partly

Reviewer #2: Partly

2. Has the statistical analysis been performed appropriately and rigorously? 

Reviewer #1: N/A

Reviewer #2: Yes

3. Have the authors made all data underlying the findings in their manuscript fully available?

Reviewer #1: No

Reviewer #2: Yes

4. Is the manuscript presented in an intelligible fashion and written in standard English?

Reviewer #1: Yes

Reviewer #2: Yes

5. Review Comments to the Author

Reviewer #1: Thank you for the opportunity to review this interesting article assessing mangers’ experiences of decision making and crisis leadership during the COVID-19 pandemic. The article is well written and easy to follow. However, I have a few questions mainly concerning the methods and use of CFIR framework.

Major issues

1. Reading the article it is unclear why you choose to use the CFIR framework. To my understanding CFIR is primarily used to guide planned and structured implementation of evidence based methods. Although it can be used for evaluation the reasoning behind choosing CFIR should be clarified.

2. The manuscript would gain from more clarity regarding what is seen as the innovation or implementation object in the present study. To me it is unclear if the innovation is the crisis management or several different innovations.

3. In the aim you state that you are exploring “how managerial decisions were made and implemented”. Please define what you mean with implemented in this case as opposed to initiated or adopted.

4. In the methods section you mention that you used purposeful sampling. Please elaborate on the sampling strategy and what kind of purposeful sampling you used (typical cases, maximum variation etc.).

5. Related to point no. 2 the analysis process needs some more explanation. For example, what was the reasoning behind the results under the heading “innovation domain”? As this domain in CFIR focuses on the qualities of the innovation (it’s evidence base, relative advantage etc.) it is hard to follow when several different innovations are mentioned.

6. The methodological discussion is very brief. I would like to see some discussions regarding preconception and trustworthiness etc.

Minor issues

1. Please clarify if the pilot interview was included in the study or not.

2. The conclusion is not very concise. Some of that is stated in the conclusion can be moved to the discussion.

Reviewer #2: 1) Research Area/Title

• The title is overly lengthy and could be more concise. Phrases like "deductive content analysis from the COPE Staff project" are redundant and detract from clarity.

• The focus on "Swedish maternity care managers" is clear, but the title does not immediately highlight the unique contribution of the study (e.g., crisis leadership strategies).

• Suggested title: “Crisis Leadership Under Uncertainty: Decision-Making Strategies of Swedish Maternity Care Managers During COVID-19”

2) Research Problems/Gap

• The gap is vaguely stated. While the abstract mentions "limited information on managers’ situation," the introduction does not explicitly critique existing literature or quantify the gap (e.g., lack of studies on maternity care managers in Sweden during COVID-19).

• The justification for focusing on Sweden (e.g., no lockdown) is buried in the introduction and should be foregrounded as a key gap.

3) Research Objective/Research Question

• The objective ("generate knowledge about managers’ regulatory strategies") is broad. A specific research question (e.g., "How did Swedish maternity care managers navigate decision-making under pandemic uncertainty?") is missing.

• The link between the CFIR framework and the research objective is unclear. Why CFIR? How does it address the gap?

4) Methodology

• The sampling strategy is not adequately justified. The rationale for selecting 18 managers from 10 hospitals is unclear, with no explanation of how data saturation was determined or whether the sample reflects diversity across roles, institutions, or regions.

• While the paper notes that the CFIR interview guide was modified, it does not provide details on how the questions were adapted or tailored to the study context. This limits transparency and makes it difficult to assess the appropriateness of the interview protocol.

• The paper states that 33 out of 48 CFIR constructs were identified in the data, but it does not explain why the remaining constructs were excluded or how their absence might influence the comprehensiveness or interpretation of the findings.

• Although the researchers’ clinical backgrounds—as obstetricians and midwives—may enrich their contextual understanding, this also introduces potential bias. However, issues of reflexivity are not addressed, and the manuscript lacks a discussion of how the researchers’ positionality may have shaped data collection or analysis.

5) Contribution to Knowledge/Practice/Methodology

• The practical contribution (e.g., "transparency in decision-making") is repetitive and lacks novelty. How do these findings differ from prior crisis leadership studies?

• The methodological contribution of using CFIR for crisis leadership is underdeveloped. The discussion should explicitly state how CFIR advanced understanding beyond generic qualitative analysis.

6) Data Analysis

• The deductive content analysis process is superficially described. Key steps (e.g., coding disagreements, inter-coder reliability) are omitted.

• Table 2 provides an example of coding but does not demonstrate rigor (e.g., frequency of codes, how themes were consolidated).

• The claim of "consensus" among coders is vague. Were disagreements documented? How were they resolved?

7) Research Findings and Discussion

• The findings section relies too heavily on direct quotes without sufficient synthesis. For example, the repeated statement that "peer support was essential" lacks deeper analysis, such as the types of support described or how these experiences varied by managerial level.

• The discussion section offers only a limited comparison to existing literature, such as Björk et al. (2022). A more thorough discussion should explicitly contrast the study’s findings with previous research on healthcare leadership during the COVID-19 pandemic.

• The limitations section notes the risk of self-report bias and the lack of employee perspectives, but it does not critically assess how these limitations may have impacted the findings. For instance, the reliance on managerial accounts could have led to an overestimation of leadership effectiveness.

8) Conclusion and Implication

• The current conclusion largely restates the results without offering clear, actionable recommendations. Implications should be specific and practical—for example: “Organizations should formalize peer-support networks to support crisis leadership.”

• The suggestion to "incorporate strategies into crisis management frameworks" is too vague. What specific frameworks are being referred to—hospital-level protocols, national emergency guidelines? And how should these strategies be incorporated—through policy updates, training modules, or structural reforms?

• Overall, the implications section is underdeveloped. It does not convincingly explain why the findings matter in a broader context, particularly beyond the COVID-19 pandemic. Given the high volume of pandemic-related research, the paper must clearly answer: Why does this study still matter now?

• While the study identifies important elements like ad-hoc decision-making, peer support, and transparent communication, it falls short of synthesizing these insights into a generalizable, transferable model for managing future healthcare crises—whether pandemics, natural disasters, or staffing shortages.

9) Recommendations for Improvement

• A dedicated subsection titled “Lessons for Future Crisis Leadership” should be added to distill the study’s findings into actionable and enduring principles for future preparedness.

• This subsection could highlight key recommendations such as implementing rapid decision-making protocols, establishing peer-support infrastructure, and developing clear internal communication channels.

• The study’s findings should be explicitly connected to ongoing healthcare challenges in the post-pandemic era, including burnout, staffing shortages, and digital transformation.

• The manuscript could explore how maternity care leaders might maintain peer networks and transparent leadership practices during non-crisis periods to enhance resilience.

• Although digital tools are mentioned in the study, their role in long-term preparedness is not fully developed. Expanding on how digital platforms can support remote coordination, triage, or staff communication would improve the study’s practical relevance.

• The discussion should be linked to global frameworks, such as the WHO Health Emergency Preparedness Framework, to emphasize the broader policy significance of the findings.

• The manuscript should also clarify the specific policy or practice gaps it addresses. The current conclusion refers vaguely to “incorporating strategies into crisis management frameworks” without specifying whose frameworks are being considered or how integration should occur.

• Providing concrete examples—such as recommending that hospital leadership training programs include scenario-based modules based on COVID-19 lessons—would significantly enhance the clarity and applicability of the study’s implications.

6. PLOS authors have the option to publish the peer review history of their article (what does this mean?). If published, this will include your full peer review and any attached files.

Reviewer #1: No

Reviewer #2: **Yes:** BAHTIAR MOHAMAD

---

## [Author Response · Author response to Decision Letter 1]

27 Aug 2025

Response to the editor and reviewers

We appreciate the opportunity to revise the manuscript with the updated title. We would like to extend our sincere thanks to the editor and reviewers for their time and for the valuable and constructive feedback, which we believe has helped improve the quality of the work. Our responses to the comments are presented below.

Comments from the Editor

Journal Requirements:

Answer: Thank you. We have ensured that the manuscript meets PLOS ONE's style requirements.

“FORMAS – the Swedish Research Council for Sustainability (2020-02767) – for funding the research”

Answer: The funders had no role in study design, data collection and analysis, decision to publish, or preparation of the manuscript. We have included the amended Role of Funders in our cover letter.

Answer: Thank you for raising this important point. We fully support open sharing of de-identified data. However, due to the extremely limited recruitment pool, large parts of the dataset includes potentially identifiable information. Nevertheless, we are able to share relevant excerpts from the transcripts upon request, pending approval from the University of Gothenburg, contact Karin.ahlberg@gu.se

Answer: We have updated the Data Availability statement in the submission form accordingly.

4. In the online submission form, you indicated that [Data contains potentially identifying and sensitive information, and there are ethical restrictions to sharing the data file as it includes individual interviews with participants. For data request, please contact The Swedish Ethical Review Authority: registrator@etikprovning.se].

Answer: Thank you for highlighting this important requirement. We fully support open science principles and the sharing of de-identified data. However, due to the highly limited recruitment pool and the sensitive nature of the interview material, the dataset contains potentially identifiable information that cannot be fully anonymized without compromising participant confidentiality. As such, public deposition would not be in compliance with the ethical approval granted for this study.

Nevertheless, we are committed to transparency and will make relevant excerpts from the transcripts available upon reasonable request, pending approval from the University of Gothenburg. Researchers interested in accessing such material may contact Professor Karin Ahlberg (Head of Department), who serves as the institutional representative for data access matters, at karin.ahlberg@gu.se.

We have updated our data availability statement accordingly and respectfully request an exemption from public deposition in line with PLOS data sharing policy.

Answer: Thank you for highlighting this important requirement. We fully support open science principles and the sharing of de-identified data. However, due to the highly limited recruitment pool and the sensitive nature of the interview material, the dataset contains potentially identifiable information that cannot be fully anonymized without compromising participant confidentiality. As such, public deposition would not be in compliance with the ethical approval granted for this study.

Nevertheless, we are committed to transparency and will make relevant excerpts from the transcripts available upon reasonable request, pending approval from the University of Gothenburg. Researchers interested in accessing such material may contact Professor Karin Ahlberg (Head of Department), who serves as the institutional representative for data access matters, at karin.ahlberg@gu.se.

We have updated our data availability statement accordingly and respectfully request an exemption from public deposition in line with PLOS data sharing policy.

Answer: Thank you. The ethics section has been moved to the Methods section.

Answer: Thank you. We have included captions for Supporting Information files at the end of the manuscript, and updated any in-text citations to match accordingly

8. Please remove all personal information, ensure that the data shared are in accordance with participant consent, and re-upload a fully anonymized data set.Note: spreadsheet columns with personal information must be removed and not hidden as all hidden columns will appear in the published file.

Answer: Due to the highly limited recruitment pool and the sensitive nature of the interview material, the dataset contains potentially identifiable information that cannot be fully anonymized without compromising participant confidentiality. As such, public deposition would not be in compliance with the ethical approval granted for this study.

Answer: Thank you for this clarification.

Reviewers' comments:

Reviewer's Responses to Questions

Comments to the Author

1. Is the manuscript technically sound, and do the data support the conclusions?

Reviewer #1: Partly

Reviewer #2: Partly

2. Has the statistical analysis been performed appropriately and rigorously?

Reviewer #1: N/A

Reviewer #2: Yes

3. Have the authors made all data underlying the findings in their manuscript fully available?

Reviewer #1: No

Reviewer #2: Yes

4. Is the manuscript presented in an intelligible fashion and written in standard English?

Reviewer #1: Yes

Reviewer #2: Yes

5. Review Comments to the Author

Reviewer #1:

Thank you for the opportunity to review this interesting article assessing mangers’ experiences of decision making and crisis leadership during the COVID-19 pandemic. The article is well written and easy to follow. However, I have a few questions mainly concerning the methods and use of CFIR framework.

Answer: Thank you for this positive feedback.

Major issues

1. Reading the article it is unclear why you choose to use the CFIR framework. To my understanding CFIR is primarily used to guide planned and structured implementation of evidence-based methods. Although it can be used for evaluation the reasoning behind choosing CFIR should be clarified.

Answer: We thank the reviewer for this valuable comment and agree that our rationale for choosing CFIR needs to be clarified. While CFIR has often been applied in planned and structured implementation efforts, it is also widely used in evaluation and can guide research across the entire process. In their systematic review, Kirk et al. (2016) highlight the value of integrating the CFIR throughout the research process – in study design, data collection, and analysis. This flexibility made CFIR particularly suitable for our study context.

During the pandemic, clinical teams could not plan their crisis management in a structured way. By applying a theoretically grounded framework such as CFIR in our study design, data collection, and analysis, we were able to capture critical dimensions of the implementation that the clinicians themselves might not have articulated. We believe that a purely inductive approach would likely not have generated the same depth and nuance. Thus, the use of CFIR allowed us to systematically explore barriers and facilitators under highly unplanned and rapidly changing circumstances, making it a valuable choice for this study.

To clarify, the following addition has been made to the methods section on page 8, line 162-167:

“ The choice of CFIR was based on its proven flexibility across study design, data collection, and analysis, as highlighted in a systematic review by Kirk et al. (23). During the pandemic, crisis management within maternity care could not be planned in a structured way. Using a theoretically grounded framework made it possible to capture important dimensions of the implementation process that might not otherwise have been articulated, thereby adding depth and nuance that a purely inductive analysis likely would not have achieved.”

2. The manuscript would gain from more clarity regarding what is seen as the innovation or implementation object in the present study. To me it is unclear if the innovation is the crisis management or several different innovations.

Answer: Thank you for this clarifying question. We have now added the following formulations to the Methods section on line 163 on page 8 to distinguish more clearly between what constitutes the innovation itself and what pertains to its implementation

“In our analysis we have defined the crisis management amongst the managers as the innovation, consisting of various measures such as decision-making, implementation of changes, and ways of managing one's own well-being”

3. In the aim you state that you are exploring “how managerial decisions were made and implemented”. Please define what you mean with implemented in this case as opposed to initiated or adopted.

Answer: Thank you for this important question. To clarify our interpretation of implementation, we have added the following formulations to the Methods section on line 155-158, page 7.

“Regarding the implementation process we consider both initiated and adopted decisions. Initiated is used interchangeably to initiated, commenced/started and launched, although they are not strictly synonymous. And implemented likewise interchangeably with adopted, approved or introduced.”

4. In the methods section you mention that you used purposeful sampling. Please elaborate on the sampling strategy and what kind of purposeful sampling you used (typical cases, maximum variation etc.).

Response: We thank the reviewer for this important comment. In the revised manuscript, we have clarified our sampling strategy on page 7 line 129-139:

“We applied a purposeful sampling strategy with the aim of achieving maximum variation. Managers were strategically selected to represent different organizational levels (first-line, middle, and senior management), both university and non-university hospitals, and geographical spread across the country. We also sought to include clinics that had made different operational choices during the pandemic (e.g., whether to allow partners on postnatal wards or not) and managers with varying lengths of leadership experience. In some cases, multiple managers from the same organization were incl

---

## [Decision Letter · Decision Letter 1]

15 Dec 2025

PONE-D-25-04207R1Crisis Leadership and Strategic Decisions in Swedish Maternity Care During the COVID-19 Pandemic: A Deductive Analysis from the COPE Staff ProjectPLOS One

Dear Dr. Stotzer,

Thank you for submitting your revised manuscript to PLOS ONE. Reviewers agree that you mostly have answered their comments and concern, but still have some minor issues. Therefore, we invite you to submit a revised version of the manuscript that addresses the points raised during the review process. Please submit your revised manuscript by Jan 29 2026 11:59PM. If you will need more time than this to complete your revisions, please reply to this message or contact the journal office at plosone@plos.org. Please include the following items when submitting your revised manuscript:

We look forward to receiving your revised manuscript.

Kind regards,

Andreas Vilhelmsson, Ph.D

Academic Editor

PLOS One

Journal Requirements:

Reviewers' comments:

Reviewer's Responses to Questions

**Comments to the Author**

1. If the authors have adequately addressed your comments raised in a previous round of review and you feel that this manuscript is now acceptable for publication, you may indicate that here to bypass the “Comments to the Author” section, enter your conflict of interest statement in the “Confidential to Editor” section, and submit your "Accept" recommendation.

Reviewer #1: (No Response)

Reviewer #2: All comments have been addressed

2. Is the manuscript technically sound, and do the data support the conclusions?

Reviewer #1: Partly

Reviewer #2: Yes

3. Has the statistical analysis been performed appropriately and rigorously? 

Reviewer #1: N/A

Reviewer #2: Yes

4. Have the authors made all data underlying the findings in their manuscript fully available?

Reviewer #1: No

Reviewer #2: Yes

5. Is the manuscript presented in an intelligible fashion and written in standard English?

Reviewer #1: Yes

Reviewer #2: No

6. Review Comments to the Author

Reviewer #1: I believe the revised version shows improvement in several areas. However, I still have some questions, particularly regarding the choice of framework and its application. You present interesting material and results that are worth disseminating, but I would encourage you to reconsider the use of an implementation framework in this study.

Below I have pasted my original comment, your answer to them and then my new comment.

Major issues

1. Reading the article it is unclear why you choose to use the CFIR framework. To my understanding CFIR is primarily used to guide planned and structured implementation of evidence-based methods. Although it can be used for evaluation the reasoning behind choosing CFIR should be clarified.

Answer: We thank the reviewer for this valuable comment and agree that our rationale for choosing CFIR needs to be clarified. While CFIR has often been applied in planned and structured implementation efforts, it is also widely used in evaluation and can guide research across the entire process. In their systematic review, Kirk et al. (2016) highlight the value of integrating the CFIR throughout the research process – in study design, data collection, and analysis. This flexibility made CFIR particularly suitable for our study context. During the pandemic, clinical teams could not plan their crisis management in a structured way. By applying a theoretically grounded framework such as CFIR in our study design, data collection, and analysis, we were able to capture critical dimensions of the implementation that the clinicians themselves might not have articulated. We believe that a purely inductive approach would likely not have generated the same depth and nuance. Thus, the use of CFIR allowed us to systematically explore barriers and facilitators under highly unplanned and rapidly changing circumstances, making it a valuable choice for this study.

To clarify, the following addition has been made to the methods section on page 8, line 162-167:

“ The choice of CFIR was based on its proven flexibility across study design, data collection, and analysis, as highlighted in a systematic review by Kirk et al. (23). During the pandemic, crisis management within maternity care could not be planned in a structured way. Using a theoretically grounded framework made it possible to capture important dimensions of the implementation process that might not otherwise have been articulated, thereby adding depth and nuance that a purely inductive analysis likely would not have achieved.”

New comment: I agree that the use of CFIR should be integrated throughout the research process. However, as Kirk et al also points out it should be used in implementation studies with identified implementation outcomes. In this study the implementation object consists of various measures and the outcome is not specified. I think using an implementation framework in a study with the aim “to generate knowledge on how Swedish maternity care managers navigated decision-making under pandemic uncertainty” is questionable as the aim is not to evaluate the implementation of an intervention.

2. The manuscript would gain from more clarity regarding what is seen as the innovation or implementation object in the present study. To me it is unclear if the innovation is the crisis management or several different innovations.

Answer: Thank you for this clarifying question. We have now added the following formulations to the Methods section on line 163 on page 8 to distinguish more clearly between what constitutes the innovation itself and what pertains to its implementation “In our analysis we have defined the crisis management amongst the managers as the innovation, consisting of various measures such as decision-making, implementation of changes, and ways of managing one's own well-being”

New comment: As for point no. 1 – do you really evaluate the implementation of a specific intervention or are you more interested in how crisis management was conducted and perceived during the pandemic (which is important and interesting, although not for implementation science)?

3. In the aim you state that you are exploring “how managerial decisions were made and implemented”. Please define what you mean with implemented in this case as opposed to initiated or adopted.

Answer: Thank you for this important question. To clarify our interpretation of implementation, we have added the following formulations to the Methods section on line 155-158, page 7.

“Regarding the implementation process we consider both initiated and adopted decisions. Initiated is

used interchangeably to initiated, commenced/started and launched, although they are not strictly

synonymous. And implemented likewise interchangeably with adopted, approved or introduced.”

New comment: What I was aiming for with my question was whether crisis leadership was actually implemented, or rather spread on its own (diffusion), and was used in different ways by different managers?

5. Related to point no. 2 the analysis process needs some more explanation. For example, what was the reasoning behind the results under the heading “innovation domain”? As this domain in CFIR focuses on the qualities of the innovation (it’s evidence base, relative advantage etc.) it is hard to follow when several different innovations are mentioned.

Answer: Thank you for this clarifying question. We have now added the following formulations to the

Methods section on line 159-161 on page 8: “In our analysis we have defined the crisis management amongst the managers as the innovation, consisting of various measures such as decision-making, implementation of changes, and ways of managing one's own well-being.”

New comment: I think the use of CFIR makes it hard to follow the results. How do you for example evaluate the evidence strength and quality, relative advantage, adaptability and complexity of crisis leadership in general? The results under “innovation domain” are more a description of how the managers tackled the situation than reflecting for example the evidence strength of the innovation (crisis leadership). Same for the other domains.

Reviewer #2: The authors have made substantial and thoughtful improvements to the manuscript, directly addressing nearly all the critical points raised in the previous review. The revisions have significantly strengthened the paper's clarity, methodological rigor, theoretical contribution, and practical relevance. The manuscript is now more focused, better justified, and positioned to make a meaningful contribution to the literature on crisis leadership in healthcare.

1. Research Area/Title

The title has been significantly improved to: "Crisis Leadership and Strategic Decisions in Swedish Maternity Care During the COVID-19 Pandemic: A Deductive Analysis from the COPE Staff Project". While the new title is more direct and includes the key concepts ("Crisis Leadership," "Strategic Decisions"), it remains somewhat long. However, it accurately reflects the study's content and is acceptable. The core issue from the previous comment (redundancy, lack of focus) has been largely resolved. The suggested shorter title could still be an option, but the current one is adequate.

2. Research Problems/Gap

The introduction now explicitly and clearly states the research gap (lines 84-99): It highlights the lack of exploration of leadership in "persistent crises" in maternity care, the scarcity of studies on Swedish healthcare managers during COVID-19, and the unique Swedish context (no lockdown). The gap is now well-defined, contextualized, and compelling. It clearly justifies why this study is necessary and novel.

3. Research Objective/Research Question

The aim is now precisely stated: "to generate knowledge on how Swedish maternity care managers navigated decision-making under pandemic uncertainty" (lines 106-107). The rationale for using CFIR is explicitly justified in the methods section (lines 100-105, 162-167). The authors explain that CFIR helps identify factors influencing implementation outcomes and, despite being designed for planned implementation, its flexibility allows it to capture important dimensions of the unplanned crisis management process. The research objective is now specific and focused. The justification for using CFIR is clear and strengthens the methodological foundation.

4. Methodology

The purposeful sampling strategy is now clearly described with a strong rationale (lines 128-139), citing "information power" and detailing the criteria for maximum variation (organizational level, hospital type, geography, operational choices, experience). Interview Guide is noted as a "modified version of the CFIR interview guide," with a test interview conducted. The Supporting Information includes the guide, enhancing transparency. The paper acknowledges that "all constructs will not necessarily be present in the data" (line 153), which manages expectations. The finding that 33 of 48 constructs were identified is presented as a result, not a flaw. Researcher Reflexivity & Bias is now robustly addressed (lines 120-127, 430-446). The authors detail their clinical backgrounds, state how this was considered during analysis, and explain that interviews were conducted by a research assistant without healthcare ties to mitigate bias. The trustworthiness section (lines 430-446) thoroughly discusses credibility, dependability, confirmability, and transferability. The methodology section is now transparent, rigorous, and defensible. The handling of reflexivity is particularly commendable.

5. Contribution to Knowledge/Practice/Methodology

A new "Methodological contribution" subsection has been added to the discussion (lines 447-463). It explicitly argues for the value of applying CFIR to crisis leadership, allowing a systematic, multi-level analysis that revealed both evident and less-articulated factors. The practical contributions (transparency, peer support, communication) are now more effectively synthesized and linked to the CFIR domains. The methodological contribution is now a key strength of the paper. The study convincingly demonstrates how CFIR can be applied beyond its traditional use.

6. Data Analysis

The data analysis section (lines 168-189) now provides a much more detailed, step-by-step account of the deductive content analysis process. It describes preparation, coding by multiple researchers, consensus-building through iterative meetings, handling discrepancies, and the consolidation of codes. It emphasizes rigor through systematic consensus-building and reflexive discussions within the multidisciplinary team. The description now meets standards for qualitative rigor, demonstrating how the analysis was conducted and how trustworthiness was ensured.

7. Research Findings and Discussion

The results are now structured logically by CFIR domain, with effective synthesis and interpretation alongside the quotes. The narrative moves beyond simple description to explain patterns and variations (e.g., different approaches to protecting pregnant staff). The discussion is substantially expanded and deepened. It now:

Engages more thoroughly with relevant literature (e.g., Björk et al., 2022; Kniffin et al., 2021) to compare and contextualize findings.

Introduces and applies the Demands-Resources model (lines 360-379) as a theoretical lens to analyze managers' experiences, which is a significant analytical upgrade.

Critically examines the implications of managers' strategies, such as the organizational vulnerability created by relying on personal time (lines 410-415).

The discussion is now analytical, theory-informed, and critically engaged with the literature. It successfully interprets the findings within a broader scholarly conversation.

8. Conclusion and Implication & 9. Recommendations for Improvement

The authors have added a comprehensive new section: "Lessons for Future Crisis Leadership Recommendations" (lines 464-516). This section is exemplary and directly addresses the core weakness of the previous version. It provides five specific, actionable lessons (rapid/revisable decision protocols, peer-support infrastructures, digital communication channels, protecting leader well-being, maintaining practices in non-crisis periods), each supported by references to the study's findings and external literature. These are distilled into four clear "Key Recommendations" (lines 507-516) for formalizing protocols, institutionalizing support, embedding digital tools, and safeguarding well-being. The conclusion (lines 517-521) is now concise and effectively points to the need to incorporate these strategies into frameworks. This is the most impactful revision. The paper transforms from a descriptive case study into a forward-looking source of evidence-based guidance for policy and practice. It answers the "so what?" question convincingly.

7. PLOS authors have the option to publish the peer review history of their article (what does this mean?). If published, this will include your full peer review and any attached files.

Reviewer #1: No

Reviewer #2: **Yes:** Bahtiar Mohamad

---

## [Author Response · Author response to Decision Letter 2]

28 Jan 2026

Response to reviewers

Answer: Thank you for the opportunity to revise this manuscript. The authors would like to thank the editor and the reviewers for the time they have devoted to this process and for their important and constructive comments, which we believe have significantly contributed to improving the quality of the manuscript.

Please find our point-to-point responses below.

Reviewer #1:

I believe the revised version shows improvement in several areas. However, I still have some questions, particularly regarding the choice of framework and its application. You present interesting material and results that are worth disseminating, but I would encourage you to reconsider the use of an implementation framework in this study.

Answer: Thank you for this positive feedback and for recognizing the value of the material and results. We appreciate the opportunity to further clarify our reasoning regarding the choice and application of the framework. We agree that this is not a typical study for the use of CFIR, and we acknowledge that this choice therefore warrants careful justification in light of your concerns.

First, we consider the use of a theoretical framework to be important in studies of this kind, as it can help capture analytical dimensions that may not be explicitly articulated by participants themselves. While inductive approaches are valuable, we judged that a purely inductive analysis might have limited our ability to systematically explore factors influencing decision-making under highly complex and rapidly changing conditions. When reviewing the emerging literature from the pandemic, we also noted that many studies were primarily observational and descriptive, with limited analytical structuring. In this context, the use of a framework offered a way to move beyond description and support analytical interpretation with potential relevance for policy and practice. We fully recognize that CFIR is not the only possible framework for such an analysis. At the time of study initiation, however, it was the most relevant framework available to us, and the urgency of securing data collection during the pandemic in order to capture real-time experiences and decision-making meant delaying data collection to explore alternative frameworks risked affecting data quality, for example through recall bias or changing organizational conditions. Our findings suggest that CFIR may have utility beyond its traditional areas of application in this specific context, although we agree that such use should be interpreted with caution.

Second, implementation processes differ substantially across fields. In clinical contexts, standardized and well-defined interventions, along with implementation processes aimed at improving patient health or clinical treatment, often represent the gold standard. In contrast, interventions aimed at improving employees’ occupational health do not follow the same logic. Instead, the importance of adapting both intervention measures and the implementation process itself has been emphasized, see for example von Thiele Schwarz, U., Nielsen, K., Edwards, K., Hasson, H., Ipsen, C., Savage, C., … Reed, J. E. (2021). How to design, implement and evaluate organizational interventions for maximum impact: the Sigtuna Principles. European Journal of Work and Organizational Psychology, 30(3), 415–427. https://doi.org/10.1080/1359432X.2020.1803960. From this perspective, the crisis management processes examined in our study can be understood as an organizational-level work environment intervention shaped by local conditions. Adaptation is therefore not a methodological weakness but a key principle in organizational workplace interventions. Applying an implementation framework such as CFIR, which has demonstrated its value in generating in-depth knowledge in evaluation studies, allowed us to systematically examine these adaptive and context-dependent processes.

Finally, the interview guide and analytical strategy were developed on the basis of CFIR from the outset. Replacing the framework post hoc and re-interpreting the data through a fundamentally different analytical structure could introduce methodological uncertainties and biases and reduce coherence between data collection and analysis, with potential implications for the study’s trustworthiness. For these reasons, we have chosen to retain the current analytical approach. In response to your comments, we have clarified our rationale and strengthened our acknowledgement of limitations in the manuscript, and we hope that these revisions address your concerns.

Below I have pasted my original comment, your answer to them and then my new comment.

Major issues

1. Reading the article it is unclear why you choose to use the CFIR framework. To my understanding CFIR is primarily used to guide planned and structured implementation of evidence-based methods. Although it can be used for evaluation the reasoning behind choosing CFIR should be clarified.

Answer: We thank the reviewer for this valuable comment and agree that our rationale for choosing CFIR needs to be clarified. While CFIR has often been applied in planned and structured implementation efforts, it is also widely used in evaluation and can guide research across the entire process. In their systematic review, Kirk et al. (2016) highlight the value of integrating the CFIR throughout the research process – in study design, data collection, and analysis. This flexibility made CFIR particularly suitable for our study context. During the pandemic, clinical teams could not plan their crisis management in a structured way. By applying a theoretically grounded framework such as CFIR in our study design, data collection, and analysis, we were able to capture critical dimensions of the implementation that the clinicians themselves might not have articulated. We believe that a purely inductive approach would likely not have generated the same depth and nuance. Thus, the use of CFIR allowed us to systematically explore barriers and facilitators under highly unplanned and rapidly changing circumstances, making it a valuable choice for this study.

To clarify, the following addition has been made to the methods section on page 8, line 162-167:

“ The choice of CFIR was based on its proven flexibility across study design, data collection, and analysis, as highlighted in a systematic review by Kirk et al. (23). During the pandemic, crisis management within maternity care could not be planned in a structured way. Using a theoretically grounded framework made it possible to capture important dimensions of the implementation process that might not otherwise have been articulated, thereby adding depth and nuance that a purely inductive analysis likely would not have achieved.”

New comment: I agree that the use of CFIR should be integrated throughout the research process. However, as Kirk et al also points out it should be used in implementation studies with identified implementation outcomes. In this study the implementation object consists of various measures and the outcome is not specified. I think using an implementation framework in a study with the aim “to generate knowledge on how Swedish maternity care managers navigated decision-making under pandemic uncertainty” is questionable as the aim is not to evaluate the implementation of an intervention.

Answer: Thank you for these clarifications to the original comment. As noted above, implementation processes differ substantially between clinical contexts—where the focus is on patient care—and organizational-level workplace interventions, which focus on employees’ occupational health. In general, managers are responsible for conducting context‑specific occupational health and safety management to prevent ill‑health among employees, in accordance with Swedish work environment legislation. This routine work is supported by internal functions such as HR departments, which possess substantial expertise in implementing work environment interventions. These resources also supported the crisis management efforts during the pandemic instructing them to perform a context‑adapted crisis management process, including designing and implementing context‑specific preventive measures to maintain operability and ensure a sustainable working environment for employees, in accordance with recommendations for organizational‑level workplace interventions.

To make this clearer, we have added the following statement to the Methods section:

“Instead, the managers were instructed to perform a context‑adapted crisis management process, including designing and implementing context‑specific preventive measures to maintain operability and ensure a sustainable working environment for employees, in accordance with recommendations for organizational‑level workplace interventions.”

In addition, after reviewing the study aim, we agree with the reviewer that it could be improved. We have revised the aim to more clearly articulate the purpose and scope of the study. The new aim reads:

“Thus, the aim of the study was to generate knowledge on how Swedish maternity care managers navigated decision‑making under pandemic uncertainty by evaluating the implementation of crisis management.”

Lastly, we fully agree that this is not a typical study for using CFIR and should therefore be interpreted with caution. To clarify, an addition has also been made to the limitation section:

“It should also be noted that the CFIR framework has most commonly been used to guide planned and structured implementation of evidence-based methods, rather than context-adjusted organisational work-environment interventions. In this study, CFIR 2.0 was applied in a context-sensitive and partial way to structure managers’ accounts of crisis management. The findings should therefore be interpreted with this specific application and context in mind.”

2. The manuscript would gain from more clarity regarding what is seen as the innovation or implementation object in the present study. To me it is unclear if the innovation is the crisis management or several different innovations.

Answer: Thank you for this clarifying question. We have now added the following formulations to the Methods section on line 163 on page 8 to distinguish more clearly between what constitutes the innovation itself and what pertains to its implementation “In our analysis we have defined the crisis management amongst the managers as the innovation, consisting of various measures such as decision-making, implementation of changes, and ways of managing one's own well-being”

New comment: As for point no. 1 – do you really evaluate the implementation of a specific intervention or are you more interested in how crisis management was conducted and perceived during the pandemic (which is important and interesting, although not for implementation science)?

Answer: Thank you for this thoughtful clarification and for engaging further with the alignment between the study aim and the use of an implementation framework. We appreciate your point regarding the importance of clearly defined implementation objects and outcomes when applying CFIR, as emphasized by Kirk et al., and we agree that this is an important consideration.

In this study, the implementation object does not consist of a single, standardized intervention but rather of a set of context-adapted crisis management measures developed and enacted by managers in response to pandemic-related uncertainty. As such, the implementation outcomes are not framed as predefined endpoints but are reflected in how these measures were initiated, adapted, and sustained in practice under rapidly changing conditions. Our intention was therefore not to evaluate the effectiveness of a specific intervention, but to examine the processes through which crisis management was implemented and navigated in an organizational context. We recognize that this positions the study somewhat different from more conventional implementation evaluations, and we have sought to clarify this distinction more explicitly in the manuscript.

In response to your comment, we have revised the study aim to better reflect this focus and to more clearly articulate that the study examines decision-making and crisis management through the lens of implementation processes, rather than evaluating a discrete intervention with predefined outcomes. We have also expanded the methodological description and limitations section to acknowledge that CFIR is most commonly applied in studies of planned and structured implementations, and that its use in the present context requires cautious interpretation.

We hope that these clarifications help to address your concerns regarding the fit between the study aim, the analytical framework, and the type of implementation processes examined, and we are grateful for your suggestions, which have helped us refine both the framing and transparency of the study.

3. In the aim you state that you are exploring “how managerial decisions were made and implemented”. Please define what you mean with implemented in this case as opposed to initiated or adopted.

Answer: Thank you for this important question. To clarify our interpretation of implementation, we have added the following formulations to the Methods section on line 155-158, page 7.

“Regarding the implementation process we consider both initiated and adopted decisions. Initiated is used interchangeably to initiated, commenced/started and launched, although they are not strictly synonymous. And implemented likewise interchangeably with adopted, approved or introduced.”

New comment: What I was aiming for with my question was whether crisis leadership was actually implemented, or rather spread on its own (diffusion), and was used in different ways by different managers?

Answer: Thank you for this clarification, which helps us better understand the intent behind your question. We agree that an important distinction can be made between crisis leadership being formally implemented, spreading through diffusion, and being enacted in different ways by different managers.

In the context of this study, crisis leadership was not introduced as a standardized or uniform model to be rolled out identically across settings. Rather, managers were tasked with carrying out a context-adapted crisis management process, which included designing, initiating, and enacting preventive and organizational measures suited to their local conditions in order to maintain operability and support a sustainable working environment for employees. This meant that crisis leadership practices were indeed applied and operationalized, but in ways that varied across managers and settings.

We therefore understand the processes examined in this study as implementation of crisis management at the organizational level, while fully acknowledging that this implementation involved substantial local adaptation and variation in how leadership practices were enacted in practice. We have clarified this distinction in the Methods section to better reflect that implementation, in this case, refers to the enactment and adaptation of crisis management measures rather than the uniform adoption of a predefined leadership model.

We hope this clarification addresses your question regarding the distinction between implementation, diffusion, and variation in leadership practices.

5. Related to point no. 2 the analysis process needs some more explanation. For example, what was the reasoning behind the results under the heading “innovation domain”? As this domain in CFIR focuses on the qualities of the innovation (it’s evidence base, relative advantage etc.) it is hard to follow when several different innovations are mentioned.

Answer: Thank you for this clarifying question. We have now added the following formulations to the

Methods section on line 159-161 on page 8: “In our analysis we have defined the crisis management amongst the managers as the innovation, consisting of various measures such as decision-making, implementation of changes, and ways of managing one's own well-being.”

New comment: I think the use of CFIR makes it hard to follow the results. How do you for example evaluate the evidence strength and quality, relative advantage, adaptability and complexity of c

---

## [Editor Report · Decision Letter 2]

22 Mar 2026

Crisis Leadership and Strategic Decisions in Swedish Maternity Care During the COVID-19 Pandemic: A Deductive Analysis from the COPE Staff Project

PONE-D-25-04207R2

Dear Dr. Stotzer,

We’re pleased to inform you that your manuscript has been judged scientifically suitable for publication and will be formally accepted for publication once it meets all outstanding technical requirements.

Kind regards,

Andreas Vilhelmsson, Ph.D

Academic Editor

PLOS One
---

## [Editor Report · Acceptance letter]

PONE-D-25-04207R2

PLOS One

Dear Dr. Stotzer,

I'm pleased to inform you that your manuscript has been deemed suitable for publication in PLOS One. Congratulations! Your manuscript is now being handed over to our production team.

Kind regards,

on behalf of

Dr. Andreas Vilhelmsson

Academic Editor

PLOS One